# Hippo Signaling Regulates High-NaCl-Induced Increase in RORγt+ Pro-Inflammatory Lymphocytes

**DOI:** 10.3390/ijms26052143

**Published:** 2025-02-27

**Authors:** Bastian Lukas Zeeb, Saskia Weber-Stiehl, Celia Escudero-Hernández, Dominik N. Müller, Andras Maifeld, Felix Sommer, Roland Schmitt, Laura Katharina Sievers

**Affiliations:** 1Institute of Clinical Molecular Biology, University Hospital Schleswig-Holstein, Christian Albrechts University, Campus Kiel, 24105 Kiel, Germany; 2Max Delbrück Center for Molecular Medicine in the Helmholtz Association, 13125 Berlin, Germany; 3Department of Internal Medicine IV, University Hospital Schleswig-Holstein, Campus Kiel, 24105 Kiel, Germany; 4Department of Internal Medicine I, University Hospital Schleswig-Holstein, Campus Kiel, 24105 Kiel, Germany

**Keywords:** hypertension, lymphocyte, NaCl, sodium, hippo pathway, Th17, TAZ

## Abstract

Arterial hypertension is a major health challenge worldwide. Lifestyle factors including dietary NaCl increase the risk of hypertension. Pathophysiologically, the activation of the renin–angiotensin–aldosterone system and vascular remodeling, as well as the increase in Th17 lymphocytes, contribute to increased blood pressure and end-organ damage. To date, it is unknown whether NaCl, changed osmolarity, and/or angiotensin II directly induce Th17 differentiation, and, if so, which molecular pathways are involved. One major transcription factor inducing Th17 differentiation is RORγt. RORγt+ immune-cell subtypes increased in a mouse model of hypertension. In primary splenocytes, NaCl and mannitol but not angiotensin II increased the frequency of RORγt+ lymphocytes and IL-17 and IL-22 expression. NaCl and angiotensin II induced angiotensin II receptor expression. NaCl led to the inactivation of the Hippo pathway in lymphocytes and decreased phosphorylation of the transcription factor TAZ, leading to increased functionality as a transcriptional coregulator. Inhibition of TAZ by verteporfin blocked the NaCl-induced increase in RORγt+ lymphocytes. Taken together, we found that NaCl induced pro-inflammatory lymphocytes via the regulation of Hippo signaling. The results suggest the possible involvement of Hippo signaling in the pathophysiology of salt-sensitive hypertension, with the potential for therapeutic targeting by small-molecule approaches.

## 1. Introduction

Arterial hypertension (AH) and chronic hypertension-related damage to the vasculature, kidneys, heart, eyes, and brain are major health challenges worldwide. Approximately 1,28 billion adults aged 30–79 years worldwide have arterial hypertension, as stated in the “World Health Organization Report 2023” [1]. Half of the concerned might be unaware of the condition, and even among patients diagnosed with AH, a large proportion are not sufficiently treated. Individuals with uncontrolled hypertension have a high risk of chronic hypertensive end-organ damage and, in turn, increased mortality. Around 20% of deaths worldwide can be attributed to the consequences of hypertension, and among individuals younger than 80 years, around 38% of overall mortality is linked to AH [1]. Traditionally, increased blood pressure has been associated with supraphysiological activation of the renin–angiotensin–aldosterone system (RAAS). Further, high dietary sodium chloride (NaCl) consumption is associated with AH. More recently, it has become clear that the immune system plays an important role in the development and maintenance of AH; for example, IL-17 secretion by T cells is a prerequisite for the full clinical extent of hypertension, most likely based on IL-17-triggered vascular dysfunction [2]. Interestingly, NaCl intake drives systemic autoimmunity via the induction of Th17 cells [3], and the NaCl-induced increase in Th17 is partly mediated by the gut microbiome [4]. Inflammatory pathways, on the one hand, perpetuate hypertension and, on the other hand, determine end-organ damage.

To date, all therapeutic approaches focus on the normalization of blood pressure, and most of them target the RAAS [5]. Further pathophysiological insight into immune cell differentiation during a hypertensive state (e.g., into the cellular signaling pathways during NaCl- and Ang-II-induced increases in Th17 cells) may contribute to the development of a therapeutic approach that targets blood pressure and end-organ damage at the same time.

Retinoid-related orphan receptor gamma (RORγt) is a critical transcription factor in T-cell fate. Its target genes IL-17, IL-23R, and CCR6 induce Th17 cells and other IL-17-secreting pro-inflammatory lymphocyte subtypes. One intracellular pathway regulating the balance between Th17 and regulatory T cells (Tregs) is the Hippo signaling pathway, which consists of a conserved kinase cascade determining the subcellular localization and regulatory activity of the transcription cofactors YAP and TAZ [6], which are paralogues with a tissue-specific expression pattern but interchangeable intracellular function. TAZ is critical for RORγt-dependent gene expression, Th17 differentiation, and tissue inflammation during autoimmune disease [6]. Depending on the interacting proteins, YAP and TAZ can have diverse effects [7]. High levels of TEA domain transcription factor (TEAD) proteins interacting with TAZ and sequestering them away from the interaction with RORγt have the same effect as TAZ knockout in lymphocytes and favor Treg differentiation [6].

Although the relevance of NaCl during AH, the contribution of the gut microbiota, and the pathophysiological implications of Th17 in AH are undoubted, it remains unclear whether NaCl and/or its downstream effector, angiotensin II (Ang II), directly affects lymphocyte function.

Hence, in this study, we set out to characterize the individual in vitro effects of key pathophysiological elements—AH, NaCl, osmolarity, and Ang II—on pro-inflammatory lymphocyte differentiation and determine the role of Hippo signaling.

## 2. Results

### 2.1. Ang II and High NaCl Increased RORγt-Positive T Lymphocytes In Vivo

A high-salt diet can cause AH, and increased dietary NaCl increases pro-inflammatory Th17 cells in vivo [3,4]. Here, we asked the following question: Which adaptations of the immune phenotype can be detected in a validated mouse model of AH in which hypertension is induced via a combination of uninephrectomy, Ang II application, and high-salt diet (Figure 1a). This in vivo mouse model resembles human AH, including hypertensive end-organ damage, in many aspects, e.g., RAAS activation and high NaCl consumption in the Western diet. After 4 weeks of chronic hypertension, the mice were sacrificed and splenic lymphocytes were examined. All mice were uninephrectomized; in addition, the hypertensive intervention group received Ang II. The percentages of T cells and B cells were similar in the hypertensive and untreated mice (Figure 1b). Further, there were no differences in subgroups, such as cytotoxic T cells (Tcs), T-helper cells (Ths), which comprise Treg and conventional T-helper cells (Tconv), γδ T cells (γδT), and innate lymphoid cells (ILCs) (Figure 1b). Typically, Th17 cells are characterized as RORγt+ and IL-17-expressing Tconv. Interestingly, there was a tendency toward increased RORγt+ cells in ILCs in the Ang-II-treated group. Sex-specific differences could neither be confirmed nor ruled out based on the low number of animals (Appendix A).

### 2.2. NaCl but Not Ang II Increases RORγt-Expressing Splenic Lymphocytes

In the next step, the effects of NaCl, iso-osmolar mannitol, and Ang II were analyzed separately in ex vivo stimulated primary mice splenocytes for 24 h (Figure 2). NaCl and mannitol increased RORγt+ lymphocytes, indicating increased proportions of Th17 cells, while Ang II had no effect. FOXP3-positive populations, as indicator of Tregs, were unchanged. TBET and GATA3, indicative of Th1 and Th2 lymphocytes, respectively, were unaltered, except at 80 mM NaCl, for which both decreased. The TBET/GATA3 ratio was constant under all conditions, indicating a stable Th1/Th2 lymphocyte balance. Taken together, the results indicate that at least part of the pro-inflammatory effect during high NaCl is elicited by the concomitant increase in osmolarity.

### 2.3. Hippo Pathway Transcription Factor TAZ Is Overexpressed in Th17 Compared to Treg Lymphocytes

Next, we asked the following question: Which genes regulate the increase in RORγt+/Th17 differentiated T cells observed during increased NaCl- and osmolarity? Thus, a pre-existing database of mouse lymphocyte gene expression was reanalyzed with a special focus on genes related to NaCl, Ang II, and hyperosmolarity [8]. We found that some typical genes involved in the control of the NaCl concentration and osmolarity, such as the epithelial sodium channel *EnaC* (*Scnn1a/Scnn1b* genes) and the osmotically regulated transcription factor *Nfat5,* were not different between naive, Th17, or Treg lymphocytes (Figure 3a). Principal components of cellular Ang-II signaling, such as angiotensin-II type 1 receptor-associated protein (*Agtrap*), angiotensin-converting enzyme (*Ace*), and angiotensinogen (*Agt*), were similarly expressed in naive, Th17, or Treg lymphocytes. Based on the hypothesis that genes differentially expressed between Th17 and naive or Treg lymphocytes might be candidates that contribute to the NaCl effect, we next systematically analyzed genes with increased Th17/naive T cell and Th17/Treg expression ratios (Figure 3b). A total of 63 genes with a ratio >2 were functionally clustered (Appendix A). Many of the genes from this list are functionally connected to inflammatory cell metabolism, ion transport, or osmotic control, which highlights the role of the extracellular composition on lymphocyte differentiation. The following six genes with an increase >5 in Th17 compared to Treg and naive T cells were found: *Gsdma, Kcnk10, Krt24, Pygo1, Sphkap,* and *Wfdc18*. For example, potassium channel subfamily K member 10 (KCNK10) is a two-pore domain potassium channel and can act as a potent modulator of osmotic volume regulation in human T lymphocytes. The osmosensor transcription factor *Nfat5* showed higher expression in Th17 and Treg than Th1 or Th2. Interestingly, the Hippo pathway transcription factor TAZ (*Wwtr1* gene) and target gene *Ctgf* showed an increased Th17/Treg and Th17/naive ratio. This is consistent with published data on the interaction between TAZ and RORγt [5].

### 2.4. NaCl Induces Hippo Pathway Activation in Lymphocytes

In the next step, we analyzed the mRNA expression of selected key signaling genes during increased NaCl, Ang II, and mannitol in vitro. In parallel with results of the FACS analyses (Figure 2), NaCl induced *RORC* and decreased *FOXP3* on mRNA levels in cultured human lymphocytes (Figure 4). NaCl increased *IL17* and *IL22* expression in a dose-dependent manner, while Ang II and mannitol had no effect on these cytokines. *IL17* increased to 268% with NaCl compared to 161% with mannitol (*p* < 0.001). The osmosensor transcription factor gene *NFAT5* was upregulated by 40 mM NaCl and mannitol. Intriguingly, *Nfat5* expression was lower at 80 mM NaCl than 40 mM NaCl. Ang II, as well as 40 mM NaCl, enhanced Ang-II receptor type 1 (*AGTR1*) transcription. *TAZ* itself and *TEAD4*, cofactor of TAZ in canonical Hippo-signaling-associated transcriptional regulation, were not significantly altered on mRNA level. Macrophage-stimulating 1 (*MST1*) was induced by Ang II, and large tumor suppressor kinase 1 (*LATS1*) increased with NaCl (Figure 4); both are kinases involved in cytoplasmic Hippo signaling. Phosphorylation by LATS kinase leads to cytoplasmic sequestration and degradation of phospho-S89 TAZ (P-TAZ), while non-phosphorylated TAZ can shuttle to the nucleus and exert their function as a transcriptional coregulator. Thus, we asked whether the observed changes in Hippo pathway gene expression were accompanied by regulation at the protein level. NaCl increased TAZ at the protein level (Figure 5a). Simultaneously, P-TAZ decreased, presumably leading to higher levels of intranuclear TAZ. Decreased P-TAZ might result from reduced activation of the upstream kinase cascade, indicated by a concomitant decrease in phospho-LATS (Figure 5a). Immunofluorescence staining of YAP and TAZ shows an almost equal distribution between the cytoplasm and nucleus, and NaCl and mannitol increased the nuclear fraction significantly (Figure 5b,c).

### 2.5. Inhibition of Hippo Signaling Abrogates NaCl’s Effect on Lymphocytes

Verteporfin is a pharmaceutical inhibitor of Hippo signaling and exerts its function via suppressing the formation of the YAP/TAZ—TEAD complex [9]. Taz was downregulated at the protein level over 24 h of treatment with 10 nM verteporfin (Appendix A). Normally, YAP/TAZ-TEAD interaction is a key component of Hippo-triggered transcriptional coregulation. Verteporfin is already approved and available for clinical use, and it is systemically administered, for example, during photodynamic therapy and treatment of macular degeneration.

Interestingly, the NaCl- and mannitol-induced increases in RORγt+ lymphocytes were blocked by verteporfin, while Foxp3-, Tbet-, and Gata3-expressing populations did not significantly change (Figure 6). Thus, we propose that NaCl-induced activation of the Hippo pathway transcription factors TAZ and YAP contribute to the induction of pro-inflammatory lymphocyte subsets during high sodium intake and possibly salt-sensitive hypertension (Figure 7).

## 3. Discussion

Building upon a mouse model of AH showing increased frequencies of RORγt+ lymphocyte subsets and human studies confirming induction of Th17 cells with a high sodium diet [3,4,10,11,12], this study shows a direct effect of NaCl on Hippo signaling, regulating murine lymphocyte differentiation in vitro, in particular, the frequencies of Th17 cells. This effect may complement the indirect regulation of Th17 cells via dietary NaCl-associated changes in the intestinal microbiome, as shown by Wilck et al. [4], and might, thus, contribute to the pathophysiology of salt-sensitive AH [10] (Figure 7).

In the mouse model of chronic hypertension, the sodium concentration in the drinking water was equal for both groups, but only the intervention group received Ang II. Traditionally, salt sensitivity in AH is understood as an exaggerated reaction of RAAS and sympathetic nervous system activation to sodium intake. Thus, it is interesting that with the separation of the complex pathophysiological stimuli ex vivo, NaCl but not Ang II induced the Th17 response. The increment of 40 mM NaCl used in the study is realistic; different conditions, such as increased salt uptake, exercise, or autoinflammatory disorders, trigger extracellular storage of NaCl in comparable or higher concentrations [11,13].

Part of the pro-inflammatory phenotype was induced by the increased osmolarity, but the IL-17 induction and RORγt+ cell increase were most potently induced by NaCl.

Importantly, this is the first study to identify an intracellular signaling pathway involved in the observed effect. NaCl and iso-osmolar mannitol apparently deactivated canonical Hippo signaling, resulting in decreased phosphorylation and impaired activation of the kinase LATS1. This fits well with a previous study proving the relevance of serum- and glucocorticoid-regulated kinase 1 (SGK1) for Th17 promotion during Ang-II-induced endothelial dysfunction and renal injury [14], since SGK1 itself is a Hippo pathway target gene. The more pronounced effect of NaCl compared to mannitol might originate, for example, from additional sodium-sensing mechanisms. The exact molecular basis of NaCl sensing was not addressed in this study; however, the “classical” sodium sensor Na_x_ (SCN7A) is expressed in lymphocytes [8], and one may hypothesize its involvement. The Hippo signaling pathway can be regulated by multiple upstream signals, e.g., extracellular matrix interactions, junction signaling, G-protein-coupled receptors, tyrosine kinase signaling, and energy metabolite availability [15]. The capacity to integrate different signals renders Hippo signaling an ideal pathway to harmonize effects (e.g., cytokine activation and the surrounding micro milieu with NaCl content, oxygen, and mechanical stiffness), which in turn might contribute to Th17 lymphocyte heterogeneity [16].

Interestingly, although T lymphocytes express Ang-II receptors, angiotensinogen, and the enzyme ACE, Ang II at physiological concentrations did not affect RORγt-, FoxP3-, Tbet-, or Gata3-positive splenocyte frequencies. However, Ang II may contribute to a pro-inflammatory phenotype during RAAS activation and AH without directly affecting Th17 frequencies. Ang II might alter lymphocyte functionality, since NaCl and Ang II induced Ang-II receptor 1 expression, and it has been shown before that Ang II alters the F-actin cytoskeleton in lymphocytes [17] and thereby modifies intercellular signaling at immunological synapses [18].

Regarding the clinical context of AH, the results suggest that effective blood pressure control (e.g., by pharmacological RAAS inhibition) might not fully ameliorate the pro-inflammatory state, which ultimately drives hypertensive end-organ damage. In addition to cardiovascular and autoimmune diseases, the complex regulation and diverse cellular effects of Hippo signaling have extensively been discussed in the fields of oncology [19,20] and anti-tumor immunology [21]. Conceivably, insight into the regulation of Hippo signaling by micro milieu effects, such as ion content and osmolarity, may aid in these fields as well.

The TAZ inhibitor verteporfin, which effectively blocked the effect of NaCl on pro-inflammatory lymphocyte differentiation, is already in clinical use for photodynamic therapy, and novel approaches utilize verteporfin-filled micro-needle patches to prevent scar formation [22]. It would be interesting to extend clinical applications to studies of the local and systemic immune phenotypes.

Taken together, our results elucidate novel mechanisms underlying salt-sensitive, low-level inflammation and lymphocyte activation. Targeting Hippo signaling in lymphocytes might be a promising approach to improve cardiovascular outcomes in patients with salt-induced inflammation and hypertension.

## 4. Methods

### 4.1. Animal Model of Hypertension and Immune Phenotyping

C57BL6 mice were bred according to the guidelines of animal welfare. The experiment comparing hypertensive mice to their untreated littermates was designed to parallel Markó et al. [23] and registered with the local authorities, LaGeSo Berlin (no. 0281/19). Initially, mice aged 7 weeks at minimum were uninephrectomized during inhalation anesthesia. After one week of recovery, arterial hypertension was induced via continuous administration of Ang II (1.44 mg/kg/day) via osmotic mini pumps (Alzet, via Charles River Laboratories, Wilmington, US) and addition of 1% NaCl to the drinking water. All mice were uninephrectomized, and the intervention group received Ang II in addition. After 4 weeks, the mice were euthanized by isoflurane overdose and, among others, the spleens were harvested for immune-cell phenotyping. A total of 50% of the mice were female, and 50% were male. Since the preliminary results showed no sex difference, the groups were combined for the analyses shown here.

Splenocyte single-cell suspensions were obtained using 70 μm strainers, followed by erythrocyte lysis and subsequent filtering using a 40 μm mesh. Cells were counted by trypan blue exclusion and labeled for flow cytometric analysis.

Isolated immune cells were either directly stained for surface markers using the respective fluorochrome-conjugated antibodies (30 min in PBS supplemented with EDTA and BSA). For all measurements, dead cell exclusion was performed using fixable viability dye for 405 nm (Thermo Fisher, Waltham, MA, USA). For intracellular staining, cells were permeabilized and fixed using the FoxP3 Staining Buffer Kit (eBioscience) and labeled using the respective antibodies. The following antibodies were used: anti-CD45-PE-Vio770 (1:100; Miltenyi, Bergisch Gladbach, Germany); anti-CD3ε-BV421 (1:50) (Becton Dickinson (BD), Franklin Lakes, NJ, USA); anti-CD4-BV711 (1:100) (BD); anti-CD8-BV650 (1:100) (BD); anti-gdTCR-BV605 (1:33) (BD); anti-FoxP3-AlexaFlour700 (1:100) (ThermoFisher); anti-RORγt-BV650 (1:50) (BD).

In this experiment, cells were analyzed with a BD FACSCanto II flow cytometer and BD FACSDiva software (version 7.0, BD Bioscience, Heidelberg, Germany). Data analysis was performed with FlowJo v10 (v.10.10; FlowJo LLC, Ashland, OR, USA).

### 4.2. Cultivation of Murine Splenocytes

For the ex vivo culture of splenocytes, C57BL6 mice were purchased from Jackson, bred according to the guidelines of animal welfare, and registered with the local authorities, MLLV (No. 1190). For the isolation of the murine splenocytes, male and female mice were euthanized, and the spleens were removed and placed in cold RPMI (Gibco/ThermoFisher, Waltham, MA, USA) +10% FCS (Biochrome/Merck, Berlin, Germany). A 70 µm sieve was washed with ice-cold PBS. Then, the spleens were placed on the sieve and gently mechanically disintegrated by pushing them through the sieve with the back of a 10 mL syringe. Afterwards, the sieve was rinsed by adding 20 mL PBS to the sample. After centrifugation (300× *g*, 6 min, 4 °C), erythrocyte lysis was performed with RBC Lysis Buffer (Invitrogen, Carlsbad, CA, USA), used as recommended by the manufacturer. The resulting cell pellet was resuspended in 1 mL RPMI + 10% FCS before the cells were counted.

### 4.3. Cell Culture

Splenocytes were seeded at approx. 10^6^ cells per 100 µL and cultured in RPMI (Gibco), +10% FCS (Biochrome) plus, and, if indicated, sodium chloride (Merck Millipore, Darmstadt, Germany), m (D)-mannitol (Roth, Karlsruhe, Germany), and angiotensin II acetate (Sigma Aldrich, Munich, Germany).

For bulk cDNA analyses, Western blotting, and immunofluorescence, a more homogenous cell population was favored, and a human immortalized lymphocyte cell line, Jurkat cells, was used under the same culture conditions as stated above.

### 4.4. Flow Cytometry

Isolated splenocytes were washed and incubated with the extracellular antibody mix for 30 min at 4 °C in the dark. Intracellular staining was performed using a commercially available FOXP3/Transcription Factor Staining Buffer Set (ThermoFisher) and incubated for 45 min at 4 °C in the dark. The viability of the cells was determined using 1:1000 PE-Texas Red Zombie Red™ Fixable Viability Kit (Biolegend, San Diego, CA, USA). The data were acquired using an SA3800 Spectral Analyzer (Sony Biotechnology, San Jose, CA, USA) and analyzed using SA3800 software version 2.0.5.54250.

The gating strategy was determined as follows: excluding doublets and non-viable cells, gating for CD3eCD4+ T-helper cells, and measuring subpopulations of Th17 (RORγt+), Treg (FoxP3+), Th1 (Tbet+), and Th2 (Gata3+) cells.

The following combinations of antigen/fluorochrome were used: CD3e/FITc (1:100) (BioLegend); CD4/Pacific Blue (1:100) (BioLegend); Gata3/PE (1:30) (ThermoFisher); RORγt/PerCP (1:100) (BD); and Tbet/PE-Cy7 (1:160) (ThermoFisher).

### 4.5. Analyses of Murine Splenocyte Expression Arrays

Preexisting microarray data on murine CD4+ T-cell transcriptomes (ArrayExpress, accession no. E-MTAB-2582) [8] were reanalyzed with regard to the expression of the genes of interest. For a hypothesis-generating overview, the ratio of the regularized logarithmic expression of all genes between Th17/Treg and Th17/naive T cells was calculated and plotted in an x–y plot.

### 4.6. Western Blot

Cell pellets from cultured cells were lysed in 100 μL RIPA lysis buffer supplemented with 1% Halt™ proteinase inhibitor cocktail (all from ThermoFisher Scientific, Waltham, MA, USA) and incubated at 4 °C for 30 min with agitation. After sonication and centrifugation at 16,000× *g* for 15 min at 4 °C, the protein supernatants were quantified using the DC Protein Assay Kit II (#5000112, Bio-Rad, Hercules, CA, USA) according to the manufacturer’s protocol.

Proteins were separated on a 10% polyacrylamide/bis-acrylamide gel and transferred to a polyvinylidene difluoride (PVDF) membrane in a semi-dry Trans-Blot Turbo™ Transfer System (all from BioRad). Blockage was performed with 5% non-fat dry milk in 0.1% Tween 20 Tris-buffered saline solution (all from ThermoFisher Scientific). Primary antibodies targeting β-actin (1:1000, ab20272, Abcam, Cambridge, UK), TAZ (1:1000, 4883, Cell Signaling Technology (CST), Cambridge, UK), phospho-Ser89 TAZ (1:2000, 59971, CST), LATS1 (1:1000, 3477S, CST), and phospho-Ser909 LATS1 (1:1000, 9157, CST) were incubated in blocking solution overnight at 4 °C. Horseradish peroxidase (HRP)-labeled secondary antibodies against rabbit or mouse immunoglobulins were incubated for 1 h at room temperature in blocking solution. The chemiluminescent signal was visualized after incubation with an ECL substrate (GE Healthcare, Freiburg, Germany) in a ChemiDoc MP Imaging System with ImageLab v6.1.0. (BioRad). All Western blot data were normalized to beta-actin. Relative quantification was performed with BioRad‘s Image Lab software Version 6.1.

### 4.7. RNA Isolation, cDNA Synthesis, and qPCR

Cells were directly lysed on 350 μL RLT buffer supplemented with 1% β-mercaptoethanol. RNA was isolated with the RNeasy Mini Kit, including DNAse digestion using the RNAse-free DNAse set (all from Qiagen, Hilden, Germany) following the manufacturer’s recommendations. RNA was quantified using the NanoDrop ND-2000 and reverse transcribed with the Maxima H Minus First-Strand cDNA synthesis kit (ThermoFisher Scientific, Waltham, MA, USA). RT-qPCR was performed using TaqMan Gene Expression Master Mix and TaqMan^®^ probes (Applied Biosystems, Darmstadt, Germany) in a VIIA 7 PCR system (ThermoFisher Scientific). Gene expression was obtained after the relative -ΔCt quantification with *ACTNB* as a reference gene (i.e., housekeeping gene). Each sample was analyzed in duplicate. The primers were designed with NCBI Primer Blast as follows:

*ACTNB* (F: GATGGTGGGCATGGGTCAG, R: CTTAATGTCACGCACGATTTC);

*AGTR1* (F: CCCCAAAAGCCAAATCCCAC, R: CCTCAAAACATGGTGCAGGC);

*IL2* (F: AACTCACCAGGATGCTCACA, R: AGCACTTCCTCCAGAGGTTTG);

*IL6* (F: ATGAACTCCTTCTCCACAAGCGC, R: GAAGAGCCCTCAGGCTGGACTG);

*IL17* (F: CCCTCAGCTACGACCCAGT, R: CTTCTGTGGATAGCGGTCCT);

*IL22* (F: TCTTGGTACAGGGAGGAGCA, R: GCCTCCTTAGCCAGCATGAA);

*FOXP3* (F: CCACATTTCATGCACCAGCTC, R: TTGAGGGAGAAGACCCCAGT);

LATS*1* (F: TGGTGTTAAGGGGAGAGCCA, R: TCCCAGCAACCCCAAGTATC);

*NFAT5* (F: GGAGAAACACGAAACGGACC, R: GAACTCCACGGGGACTGATT);

*RORC* (F: CGGGCCTACAATGCTGACA, R: GCCACCGTATTTGCCTTCAA);

*TEAD4* (F: CCGGCACCATTACCTCCAAC, R: CTGCTCAATATCCGGGCTCC);

*WWTR1* (F: ACATGGCTACAAGGACCAGG, R: TTCGAGGTGCCAGAGCAAAT);

*YAP1* (F: CCCTCGTTTTGCCATGAACC, R: GTTGCTGCTGGTTGGAGTTG).

The relative transcript levels were determined using *Actb* as the housekeeper.

### 4.8. Immunofluorescent Staining

For the staining, Jurkat cells were fixed for 10 min at 37 °C in 4% paraformaldehyde in phosphate-buffered saline (PBS) after 24 h of stimulation, as described above. Cells were blocked at room temperature for 15 min in 0.1% Triton-X-100 (Sigma Aldrich) buffered in PBS, followed by 1 h in 2% BSA in PBS. Staining was performed using YAP/TAZ antibody (1:100 in 0.1% BSA, 8418S, CST) for 1.5 h at room temperature. Next, cells were washed in PBS, followed by incubation with a secondary antibody, nuclear staining, and cytoskeletal staining with Alexa Fluor 488-labeled goat anti-rabbit IgG (diluted 1:500 in 0.1% BSA (ThermoFischer, A11034)) with DAPI (1:40,000 in PBS, D9542, Sigma Aldrich) and Rhodamine Phalloidin (1:400, T1162, Sigma Aldrich) for 45 min at room temperature. The cells were then air-dried on a microscopy slide and mounted using antifade mounting media (DAKO, Hovedstaden, Denmark). Images were captured using an imager Z1 microscope (Zeiss, Jena, Germany) and analyzed with the ImageJ2 open-source processing software (version 2.14.0/1.54f).

### 4.9. Graphic Representation and Statistics

Graphs were plotted and statistical analyses were conducted using the GraphPad Prism 10 software package (GraphPad Software, San Diego, CA, USA). The specific statistical test employed is indicated in the figure captions. *p*-Values smaller than 0.05 were considered statistically significant; correction for multiple testing was not performed based on the small number of pre-planned comparisons.

## Figures and Tables

**Figure 1 ijms-26-02143-f001:**
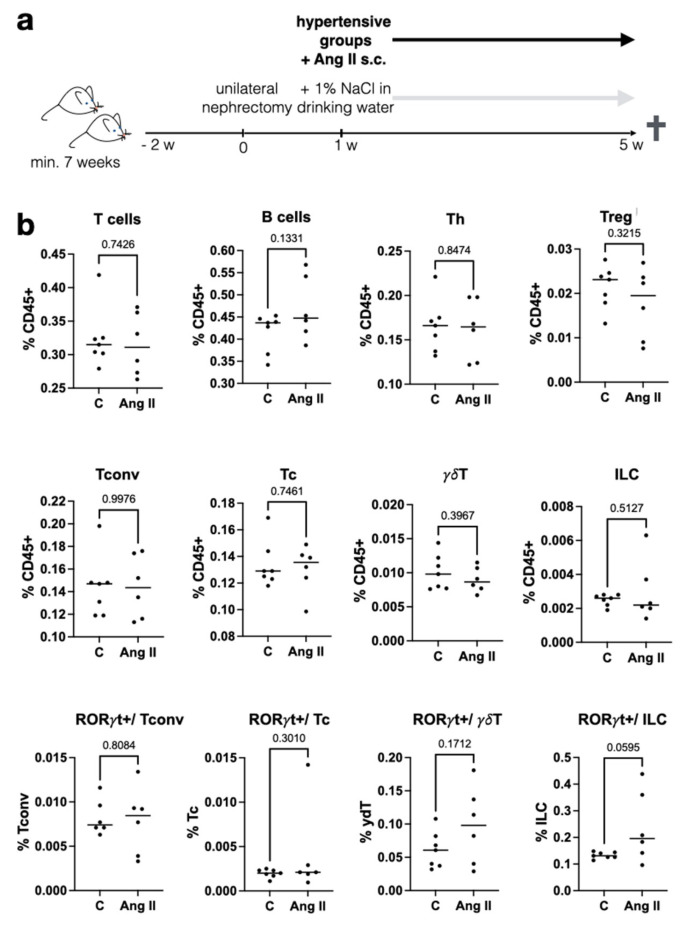
Immune cell subtypes were stable in an in vivo model of hypertension but show increased RORγt+ lymphocytes. (**a**) For the induction of chronic hypertension, mice were subjected to uninephrectomy, implantation of osmotic Ang-II mini pumps, and high dietary NaCl. Male and female mice were pooled, and the numbers varied by differences in survival. The control group consisted of 3 female and 4 male mice; in the Ang-II group, 2 were female and 4 male. (**b**) Spleens were lysed and analyzed via flow cytometry: T cells (CD3+), B cells (CD45+, CD3-, CD19+), cytotoxic T cells (Tc; CD3+ gdTCR- CD4- CD8+), T-helper cells (Th; CD3+ γδTCR- CD4+ CD8-), Treg (CD3+ γδTCR- CD4+ CD8- FOXP3+), conventional T-helper cells (Tconv.; CD3+ γδTCR- CD4+ CD8- FOXP3-), and innate lymphoid cells (ILCs; CD45+ CD3-,TBET+ RORγt+ GATA3+). Each dot denotes one animal. Statistical test: unpaired, two-tailed *t*-test (*p*-value given).

**Figure 2 ijms-26-02143-f002:**
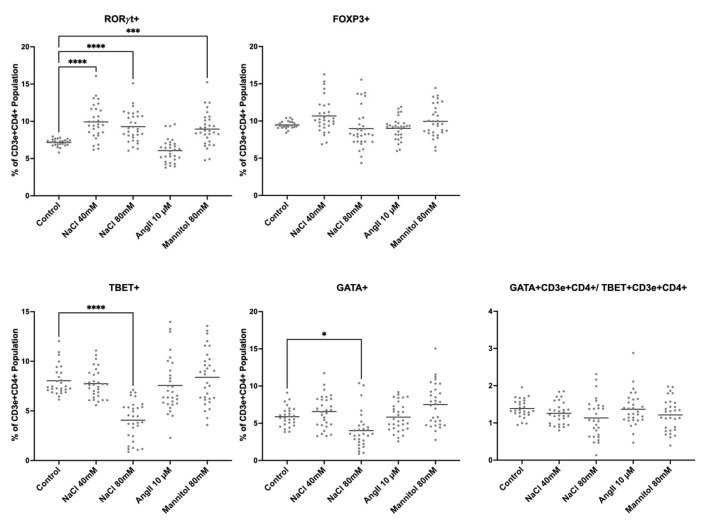
NaCl induced CD3e+CD4+RORγt+ lymphocytes. Flow cytometry of primary cultured splenocytes that were treated for 24 h with NaCl, angiotensin II (Ang II), and mannitol and stained for the main lymphocyte marker proteins: RORγt (e.g., Th17 cells), FoxP3 (Treg), Tbet (Th1), and Gata3 (Th2). Each dot denotes data from one spleen culture. n = 31; statistical test: one-way ANOVA; ****, *p* < 0.0001; *** *p* < 0.001 and * *p* < 0.05.

**Figure 3 ijms-26-02143-f003:**
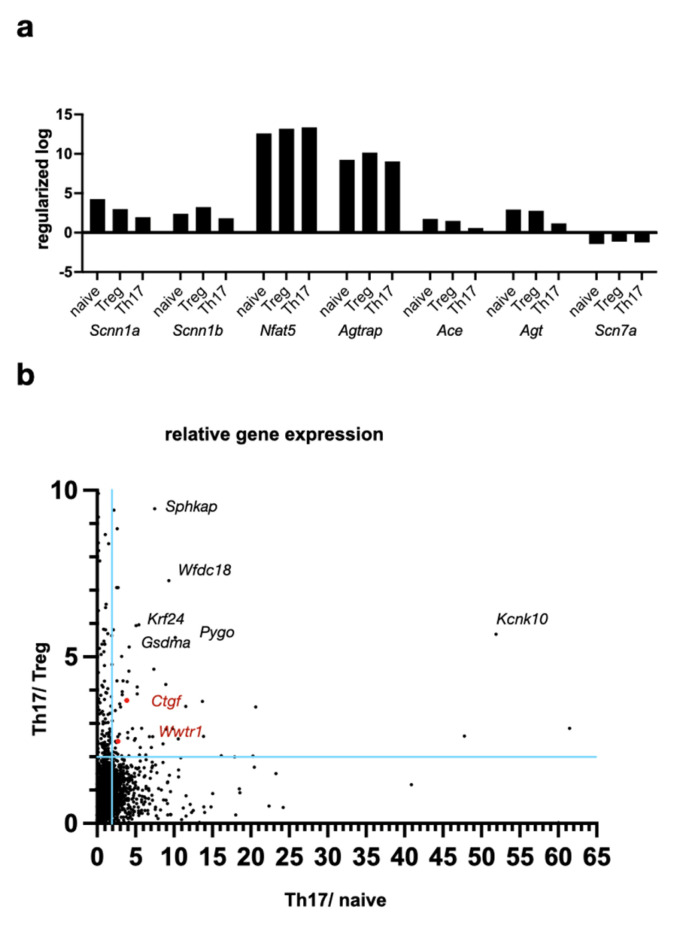
Hippo signaling components TAZ and CTGF were expressed higher in Th17 than naive and Treg: (**a**) regularized logarithmic gene expression of selected genes in murine T-lymphocyte subtypes; (**b**) ratio of the regularized logarithmic expression of all genes between Th17/Treg and Th17/naive T cells. Each dot depicts one gene. The *Wwtr1* gene, which codes for the TAZ protein, and the TAZ target gene *Ctgf* are marked in red (*Wwtr1*: 2.36/2.26; *C*tgf: 3.77/3.69; all data provided in Appendix A) [7].

**Figure 4 ijms-26-02143-f004:**
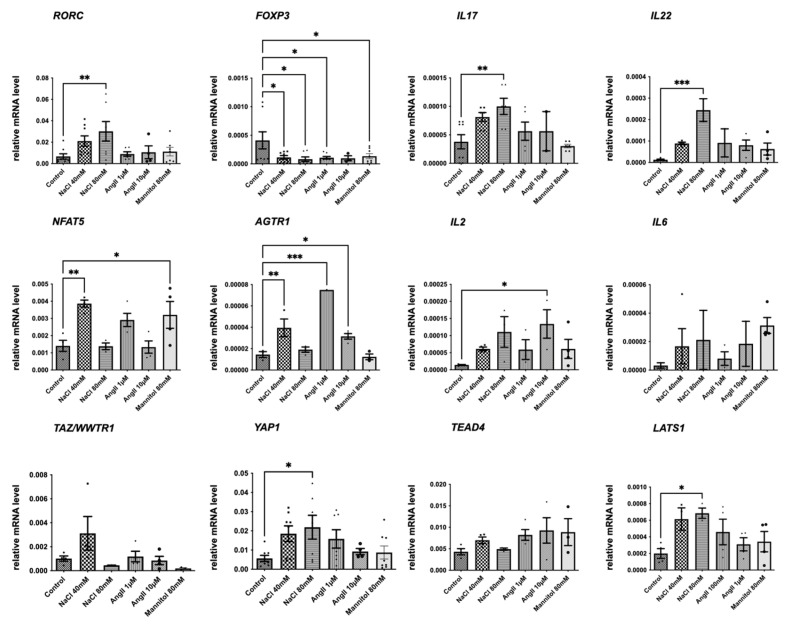
NaCl induced proinflammatory cytokines. mRNA expression analyses via qPCR of human immortalized lymphocytes cultured under the indicated conditions for 24 h; n = 4. statistical test: one-way ANOVA; *** *p* < 0.001, ** *p* < 0.005, and * *p* < 0.05.

**Figure 5 ijms-26-02143-f005:**
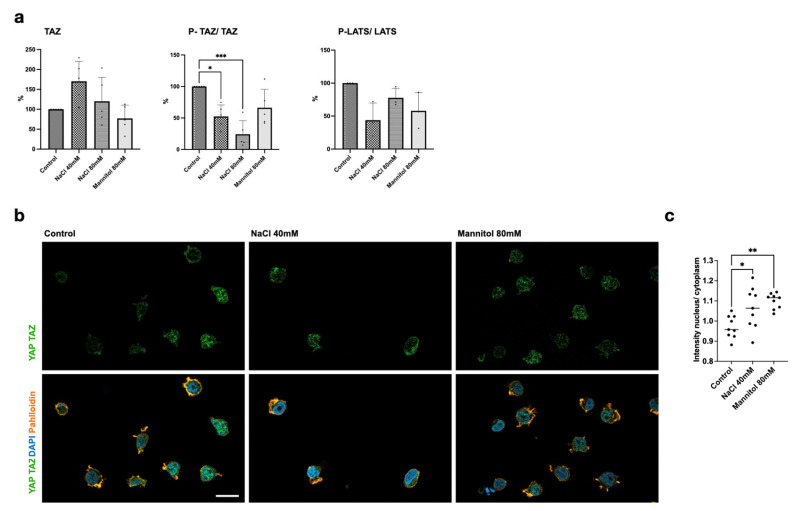
NaCl induced phosphorylation and increased nuclear localization of TAZ in lymphocytes. (**a**) Semiquantitative Western blots of protein expression from human immortalized lymphocytes. n = 6; statistical test: one-way ANOVA; *** *p* < 0.001, ** *p* < 0.005 and * *p* < 0.05. (**b**) Representative immunofluorescence staining of TAZ and YAP (green) in primary cultured splenic lymphocytes treated with NaCl or mannitol for 24 h. Cytoskeleton was contrasted (red) and nuclei were stained with DAPI (blue). The scale bar denotes 20 µm. (**c**) The fluorescence intensity ratios of YAP/TAZ in the nuclei and cytoplasm of 10 arbitrarily chosen cells were calculated. n = 10; statistical test: one-way ANOVA; ** *p* < 0.005 and * *p* < 0.05.

**Figure 6 ijms-26-02143-f006:**
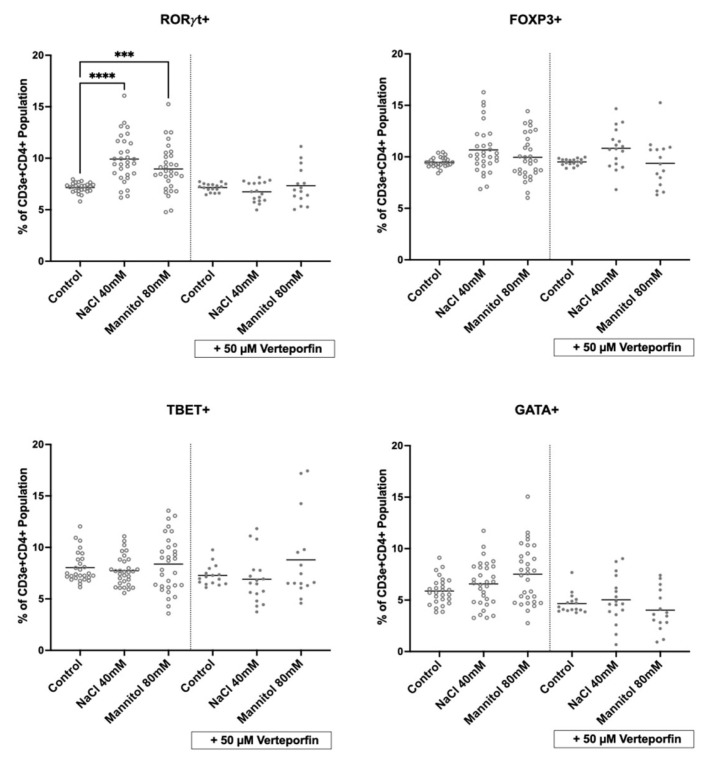
Inhibition of TAZ by verteporfin blocked the NaCl-induced increase in CD3e+CD4+RORγt+ lymphocytes. Flow cytometry of primary cultured splenocytes treated with NaCl or mannitol for 24 h in the presence or absence of verteporfin, an inhibitor of TAZ, and stained for the main lymphocyte marker proteins RORγt (e.g., Th17 cells), FoxP3 (Treg), Tbet (Th1), and Gata3 (Th2). n = 31, for the control; n = 17, for verteporfin. Statistical test: one-way ANOVA; **** *p* < 0.0001 and *** *p* < 0.001.

**Figure 7 ijms-26-02143-f007:**
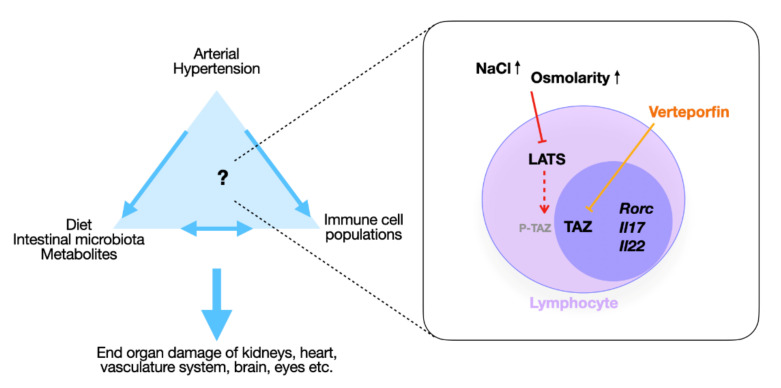
Hippo signaling regulates high-NaCl-induced increases in RORγt+ pro-inflammatory lymphocytes. Schematic overview of the results.

## Data Availability

All data are contained within the article.

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
