# Peer review of "Hippo Signaling Regulates High-NaCl-Induced Increase in RORγt+ Pro-Inflammatory Lymphocytes"

_ijms, 2025, doi:10.3390/ijms26052143_

Round 1

Reviewer 1 Report

Comments and Suggestions for Authors

Zeeb and colleagues explored the role of Hippo signalling in the induction of pro-inflammatory lymphocytes, particularly Th17, in the setting of salt-sensitive hypertension. They have employed a range of in vivo and in vitro techniques, along with analyses of online databases to generate data for this manuscript. There are some interesting findings, that could possibly be enhanced after addressing the following comments:

Major comments

1.         Line 84-85: “there was a tendency of increased RORgt cells in different immune cell subtypes including ILC, gdT and Tconv in the Ang II treated group”. Given the p values in Figure 1b, other than the RORgt+/ ILC subset, this is statement is not valid. Furthermore, how was the statistical power determined?

2.         Results sections 2.3 & 2.4: It is unclear whether human cells/ mouse lymphocytes were used to obtain these results. Figures legends could be improved by clearly stating the cell type and nature or the experiment

3.         In Figure 2, 40mM NaCl caused an increase in the proportion of CD3e+CD4+RORgt cells, yet there was no relative reductions in FoxP3+, Tbet+ and GATA3+ populations. Presumably if there was a relative increase in one population, I would expect there to be a reduction in others given the expansion if relative to the CD3e+CD4+ population. Is the data actually reporting cell counts or there is a large proportion of naïve CD3e+CD4+ not expression the relevant transcription factors?

4.         How do the authors reconcile the lack of a concentration-dependent relationship between salt and Th17 polarization? 

5.         Figure 3b is an interesting set of data. While it is understandable that the focus was placed on Taz and CTGF, there are many other upregulated genes that are represented in the high Th17:Treg and Th17:naive quadrant. What are these genes and could they be of relevance to the results obtained in this study? The authors should expand on these findings in the results and discussion. 

6.         Figure 3b: The size of this figure could be expanded and/or dot size and annotations.

7.         Figure 4b: There appears to be a reduction in the nuclear Taz/Yap in response to 40mM NaCl. While the low power images showing multiple cells is important to represent the heterogenous response, higher power images would be required to show the nuclear translocation. There is clearly less nuclear Taz/Yap compared to the control. 

8.         Given that a model of hypertension was used, was the blood pressure measured and confirmed where animal studies were conducted?

9.         The control mice used for section 2.1 are maintained on 1% NaCl in addition to being uninephrectomised. It would be important to report if there was an effect on splenic T and B cell subsets as a result of uninephrectomy and 1% NaCl in control mice? This is important since the following results of the manuscript confirms a role for NaCl (and not Ang II) in Th17 induction. Were experiments performed on naïve control mice with uninephrectomy and 1%NaCl?

Minor comments

Introduction

1.         Line 33: Mention some stats at the start of the introduction. Global prevalence, mortality rate etc.

2.         Line 50: define what RORgt stands for 

3.         Line 58: define what TEAD stands for

Results

1.         Line 75: remove the “?” after “(Figure 1a)”

2.         Line 79: “The count of T cells and B cells…” yet the figure shows immune cell % s.

3.         Line 99: Perhaps introduce Tbet and Gata3 beforehand in the main text.

4.         Figure 2: No Y-axis title in the final graph

5.         Line 108-109: Title is not clear, please reword

6.         Line 113: EnaC not Enac

7.         Lines 110-122: Wording on this paragraph is not clear

8.         Figure 3A: Do you need the genes you don’t talk about? AGTRAP, ACE, AGT and SCN7A

9.         Line 136-137: Sentence is not grammatically correct

10.   Results sections 2.3 & 2.4: Please use the correct gene nomenclature (eg: Il17, Il22, Il6 when referring to rodent genes and IL17, IL22, IL6 when referring to human genes)

11.   Results section 2.4: Be more specific with the NaCl data as 2 different NaCl concentrations have been used

12.   Line 135: Did mannitol increase Il17 expression? In line 133 it is mentioned that mannitol didn’t have an effect on Il17

13.   Lines 138-139: Introduce MST1 and LATS1 briefly

14.   Line 134 and figure 4: Il6 is not increased in the figure, yet the results say NaCl induced Il6?

15.   Results section 2.4/ Figure 5b: mention that immunofluorescent stained cells are the Jurkat cells, higher magnification is required for the images

Methods

1.         Line 243: What’s the sex of the mice used for the animal studies?

2.         Line 248: Is the Ang II dose 1.44 mg/kg/day?

3.         Section 4.2: Were there any sex differences given that both male and female mice have been used for ex vivoculture of splenocytes? No mention in the results or discussion

4.         Section 4.7, line 334:  what’s the housekeeping gene used? Is it Actb?

Other

1.         Add a dash between IL and 17 (IL-17, not IL17), same goes for the other interleukins mentioned

2.         Perhaps include representative flow cytometry plots and westerns alongside the quantified data

3.         Abbreviations list is incomplete (no mention of interleukins, MST1, ILC etc.)

Comments on the Quality of English Language

The quality of the English is satisfactory.

Reviewer 2 Report

Comments and Suggestions for Authors

In the manuscript of Zeeb et al., the influence of NaCl induced hypertension regulation via Th17 and other T cells and Hippo signaling is recomended.

In the introduction the authors describe their study . They want to characterize the individual in vitro effects of the pathophysiology of AH, NaCl on lymphocyte differentiation.

The results are very difficult to follow. There is an in vivo experiment with uninephroectomy. Why is it necessary to remove the kidneys.? 

There were no effect in splenic lymphocyte populations in this experiiments.

How high is the NaCl concentration at the end in the plasma of mice?

In the next ex vivo experiment splenocytes  of (untreated?) mice were stimulated with extremly high and unphysiological amounts of NaCl . What are "normal" and "high salt " concentrations in vivo possibly inducing pathogenic Th17 T cells, which is already published in several studies. 

It would be nice to see the original western blot in Figure 5 a, there is only the quantification of the westernblot visible

I cannot understand the conclusions of the experiments. There is a down regulation with verteporfin in ROR gamma T cells and Gata+ T cells .The inhibition of Taz is not shown.

All in all I think that there are some data missing for this conclusion. 

There are different effects in different salt concentrations , this is sometimes contradictory.

Comments on the Quality of English Language

Parts of the result section and discussion have to be rewritten,

Please correct part 2.1 results . There are some clear and understandable sentences missing.

e.g. AngII application and high salt diet (Figure 1a)? 

Round 2

Reviewer 1 Report

Comments and Suggestions for Authors

I am satisfied with the responses.

Author Response

Thank you very much!

Reviewer 2 Report

Comments and Suggestions for Authors

Dear authors , thank you for the point by point reply.

I have some additional concerns regarding some data:

In your point by point reply   one graph was added, demonstrating the Na levels in serum and total urinary. There is a high difference ,  but no significant difference in splenocytes in ROR gamma T -T cells. In vitro the  effect is much more clear. It seems that there is a very high variation in mice, therefore the amount of mice (n=6?). In the reveised version you mentioned that 50% of the 6? mice were male and 50% were female?  What does that mean for the outcome?

Why did the authors use two tailied T test in figure 1 and not Man-whitney U test (seems not to be normal distribution? )

There are double graphs in figure 2. Please describe, that splenocytes from naive mice are used.

Thank you for submitting the original western blot data.

The data in Figure 5 show an increase of TAZ with 40mM NaCl (n=5), nevertheless the controls do not show any standard deviation, although the blots show different expression levels of TAZ in the controls.  Are the expression levels normalized to beta actin? I dont think so, there are different beta actin expression levels in some blots (e.g. BZ023)! BZ18, controls, TAZ expression, there are two control lanes, differently expressing TAZ (beta actin seems in this case relatively homogenous!). I think the western blot quantifications are not normalized to house keeping proteins and are of very low quality, I am not convinced that TAZ is significantly upregulated with 40mM NaCl, there are also blots showing low TAZ expression with higher and lower NaCl concentrations.

Although verteporfin is published as an inhibitor, there are no experimental data, that TAZ /Yap are reduced or blocked in your system, there is a clear immunofluorescence or western blot missing that there is a downregulation, otherwise the FCM data are only speculation.

Therefore I am not convinced with the model proposed since, in my opineon the experimental data are not clearly demonstrated.

Round 3

Reviewer 2 Report

Comments and Suggestions for Authors

Dear authors,

thank you for the additional data. Please include in the final manuscript the data you sent to me as additional figure files. Please describe the number of animals in the legend and that the animals were pooled data from male and female( how many?, the number is rather low!)

Pleas mention how you normalized the western blots ( you did not mention in the Material and methods) and always write the numbers in the legend.

Author Response

Dear reviewer,

thank you again for your thorough review and constructive feedback! We have optimized the manuscript according to your suggestions.

Laura Sievers